# Assessment of Balance Parameters in Children with Weakened Axial Muscle Tone Undergoing Sensory Integration Therapy

**DOI:** 10.3390/children10050845

**Published:** 2023-05-07

**Authors:** Jadwiga Jacewicz, Alicja Dziuba-Słonina, Agnieszka Chwałczyńska

**Affiliations:** 1Department of Physiotherapy in Neurology and Pediatrics, Wroclaw University of Health and Sport Sciences, 51-612 Wroclaw, Poland; alicja.dziuba@awf.wroc.pl; 2Department of Human Biology, Wroclaw University of Health and Sport Sciences, 51-612 Wroclaw, Poland; agnieszka.chwalczynska@awf.wroc.pl

**Keywords:** sensory integration therapy, equilibrium parameters, Zebris platform, weakened axial muscle tone, holistic rehabilitation

## Abstract

Children with weakened axial muscle tone face various problems every day. One is maintaining a stable body posture, which limits their participation in activities and games with peers. The study aimed to assess balance parameters in children with weakened axial muscle tone who underwent sensory integration therapy (SI). The study group consisted of 21 children (divided into three age groups) referred by a doctor for therapy. Methods: The ZEBRIS platform was used to measure the balance parameters (MCoCx, MCoCy, SPL, WoE, HoE, and AoE). The study was conducted twice: before and after two months of sensory integration therapy. The results were compiled using the TIBICO^®^ Statistica software version 13.3.0. Results: After the SI program, statistically significant changes were observed in the values of MCoCy_oe, WoE_oe, AoE_oe in the group of four-year-olds, MCoCX_ce in the group of five-year-olds, and in SPL_ce and AoE_ce in six-year-olds. A statistically significant, highly positive correlation was observed between body height and changes in SPL_oe, HoE_oe, and AoE_oe in the group of six-year-olds, as well as in the case of changes in SPL_oe in the group of five-year-olds. In the group of four-year-olds, a statistically significant correlation occurred only between body height and the change in the MCoCx_oe value. Conclusions: the sensory integration therapy used in the study group of 4–6-year-old children with reduced muscle tone gave positive results in the form of improved static balance and balance.

## 1. Introduction

The development of a child, although determined by certain standards adapted to age, depends on the environment in which the child is brought up as well as on his health. The latest data suggest that even a properly developing child needs systematization of data from developing senses, such as taste, smell, sight, hearing, movement, gravity, and body position in space, i.e., the use of sensory integration (SI). Many authors point out that 10–55% of physically healthy children have SI disorders, and in the case of developmental abnormalities or chronic diseases, this percentage increases to 88% depending on the type and degree of dysfunction [1,2,3,4]. The correct integration of stimuli received from the external and internal environments in children affects their participation, especially in activities of daily living (ADL), such as dressing, eating, moving, engaging in play, and participation in recreational and school-related activities, as well as instrumental activities of daily living (IADL) [1,5]. Disturbances in ADL and IADL may be caused, among other things, by reduced muscle tone. Children with weakened muscle tone often have problems with properly integrating stimuli because skeletal muscles (whose strength is weakened in these children) are involved in mobility, strength, and balance [6], enabling participation and performance of various activities. Previous studies have shown that low muscle mass and strength contribute to adverse health effects in childhood, such as increased risk of metabolic dysfunctions, cardiovascular diseases, reduction of bone minerals or poorer cognitive and motor performance in early childhood compared to other children of the same age [7,8,9,10,11,12,13,14,15,16]. Weakened muscle tone may include neurological, genetic, or idiopathic causes and applies to such disease entities as cerebral palsy [17,18], Down’s syndrome [19], and autism spectrum [20], as well as being associated with various birth defects, e.g., central nervous system. These children in the later period struggle not only with motor difficulties but also with cognitive ones.

The current goal of sensory integration therapy (SIT) is to reduce or eliminate problems of neuronal disorganization. Neuronal disorganization refers to a problem in responding appropriately to the stimuli reaching the body. Sensory systems include proprioception from muscles and joints, balance and hearing from the vestibuloauditory system, visual stimuli provided by vision, tactile sensory stimuli from the skin, and body awareness [21]. One of the four areas in sensory integration is postural control, which is based on two systems: vestibular and proprioceptive. The vestibular system provides sensory information that helps in orientation and perception of movement, balance, locomotion, and stabilization of eye movements [22]. Its primary functions are regulating the state of arousal and regulating muscle tone (i.e., motor function). Proprioception is a component of feeling the position of one’s body parts and the level of effort exerted by the working muscles, so it is an essential part of movement and posture control. In the human musculoskeletal system, the two main types of mechanoreceptors involved are the muscle spindle and the Golgi tendon organ (GTO) [23,24,25].

The convergence of proprioceptive afferent cervical fibers with vestibular and visual stimuli at different levels of the neural axis involves the vestibular nuclei, thalamus, and cerebral cortex [26,27,28,29,30,31,32]. Therefore, integrated work and visual–vestibular and proprioceptive–vestibular interactions are crucial for postural control [26,33]. Balance is achieved in an active way by selecting and engaging the work of appropriate muscle groups and their activation at the right time [34]. When information from different sensory systems is inaccurate and/or conflicting, the central integration of inputs from the visual and vestibular sensory systems can stabilize the center of mass (COM) [35] and, thus, prevent falls. Hence, the central integration process allows us to choose a specific response strategy to maintain postural control according to external posture, displacement, and previous experience [36,37].

Appropriately initiated SIT is effective in the case of classic forms of sensory integration disorders in diseases such as attention deficit disorder, mental retardation, autism spectrum (in which sensory dysfunction is associated with disturbances in the modulation area) [38,39], and genetically determined syndromes and plays the role of supportive therapy in them [40,41]. The available literature confirms that SIT also has positive effects in such aspects as participation in activities in the case of children with autism spectrum disorder [5,42], improves concentration in children with left-sided infantile hemiplegia [21], and reduces the stimulation of the central nervous system [43]. To quote Pyda-Dulewicz and co-workers: “…Sensory Integration Techniques, by influencing the vestibular system and the sense of deep sensation, can be an alternative or, in the case of older children, a supplement to standard vestibular habituation training in the treatment of balance disorders” [44].

Therefore, the work areas during SIT coincide with those needed to maintain balance. This is an integral work of the vestibular, proprioceptive, and vision systems. The aim of the study was to investigate whether SI has an impact on balance parameters in children with weakened muscle tone. Bearing in mind the basics of SIT, the question arises whether it can be used in the case of balance training in children with weakened axial muscle tone and whether the age of the examined person has a significant impact on the effects of the applied therapy.

## 2. Materials and Methods

The research received the consent of the Research Ethics Committee at the University of Physical Education in Wrocław No. 8/2016 of 21 February 2016. The research was carried out in accordance with the Declaration of Helsinki. The written consent of the parents/legal guardians of the examined children was obtained for participation in the research project. At the same time, the child’s consent to participate in therapy was obtained each time during the classes.

### 2.1. Study Group

The condition for inclusion in the study group was a medical diagnosis made by a specialist doctor—a rehabilitation doctor and/or a neurologist—indicating the presence of reduced axial tone of skeletal muscles; ability to independently assume and maintain a standing position; age between 4 and 6 years. Examination by a specialist included observation of spontaneous motor skills in various positions and examination of reflexes and tension. In Poland, the main scale on the basis of which the level of muscle tone is diagnosed is the modified Ashworth scale. The research project did not include children with a history of injuries, procedures, or operations in the last 6 months before the start of therapy and significant intellectual disability manifested by the lack of contact with the therapist. Children who participated in SI classes regularly (attendance >90%) once a week for a period of 10 weeks qualified for the study group.

### 2.2. Experiment Description

The study was conducted in a children’s rehabilitation clinic in Wrocław. The participant was informed about the course of the experiment. The tests were carried using the ZEBRIS PDM-S Platform, the sampling frequency of which is 200 Hz. Since the test group consisted of children, each participant stood on the platform before the actual test to become familiarized with the surface. The measurement was performed without footwear and outer clothing of the feet. The first measurement was a static measurement—eyes open and eyes closed—during which the load on the limbs while standing was assessed. The second measurement was the equilibrium analysis measurement. It consisted of standing for 30 s, first with eyes open and then with eyes closed. In the second measurement, there was a sufficiently long break between each measurement (approximately 30–60 s). The children were examined twice: at the beginning and after two months of the sensory integration therapy—which they attended once a week for this period of 10 weeks. Each class lasted 40 min.

One therapy session included the following tasks:Vestibular stimulation—rotational movement: rolling in a barrel through sensory berets.Proprioceptive stimulation and work on concentration—squeezing in a quadruple position between two large gymnastic balls with an attention task, pushing with the hands from the ground while lying face down on a skateboard with an attention taskVestibular stimulation—front–back linear movement: swinging while lying face down on a swing with the hands holding the thera-band attached to the ladders,Balance exercises, active sensory stimulation of the feet—overcoming an obstacle course made of sensory berets and doormats with different texturesPassive stimulation of deep sensation, calming down—joint pressures, pressure massage with a gymnastic ball.

The following is a list of the analyzed parameters from the equilibrium analysis measurement and their abbreviations:-Load on the left side—load on the left lower limb during free standing expressed in % of the total load (both limbs);-Load on the right side—load on the right lower limb during free standing expressed in % of the total load (both limbs);-MCoCx (cm)—X (in the frontal plane) coordinate of the mean value of CoP (center of pressure), based on the point of origin of the platform;-MCoCy (cm)— X (in the sagittal plane) coordinate of the mean value of CoP (center of pressure), based on the point of origin of the platform;-SPL (cm)—the sway path length (length of way of CoP);

Inside the calculated ellipse is 95% of the measurement data of the location of CoP:-WoE (cm)—amplitude;-HoE (cm)—height of amplitude;-AoE (cm^2^) is the area of the ellipse (AoE indicates the orientation of the direction of the longitudinal axis of the ellipse compared to the longitudinal axis of the platform (indicating clockwise rotation).

### 2.3. Statistical Analysis

Comparison of the results obtained before and after the therapy was made using the Statistica program, version 13.3.0. The “oe” marking next to a given variable means that the measurement was made with open eyes, and the “ce” marking indicates a result obtained with closed eyes. Descriptive statistics, such as the mean and standard deviations, were used to describe the groups. On the basis of the Shapiro–Wilk test, the lack of normality of the distribution of the examined features in particular groups depending on the age of the examined children was determined. The nonparametric Kruskal–Walli test statistics were used for comparison among groups. The assessment of the changes in the parameters of the static equilibrium was carried out using nonparametric tests for dependent groups—the Wilcoxon pairwise order test. The relationship between anthropometric values and changes in the static balance parameters was assessed using Spearman’s rank correlation. The level of statistical significance was *p* < 0.05.

## 3. Results

The study group consisted of 21 children aged 4–6 years. The subjects were divided into three groups according to age—group I (*n* = 7) of children aged 4, group II (*n* = 7)—5-year-olds, and group III (*n* = 7)—6-year-olds. Qualification to the study group is presented in the CONSOR diagram (Figure 1).

When evaluating the balance parameters in children before the start of SIT, no statistically significant differences were observed in the results obtained between the five- and six-year-olds and the four- and six-year-olds. Statistically significant differences were observed in the study of balance in the sagittal plane with eyes open (MCoCy_oe) between the examined 4-year-olds and 5-year-olds; the other values in these groups did not differ. The results obtained before the start of the SIT showed that 4-year-olds had the smallest mean sideways and anterior–posterior deflections relative to the support plane compared to the 5- and 6-year-olds, both when measured with eyes open and closed. The same situation concerned the length of the CoP point path during the study but only when measured with eyes open—the mean distance was the shortest in the case of the 4-year-olds, while when measured with eyes closed, the 4-year-olds had the longest mean distance of the CoP (more than twice as long as 6-year-olds). In the case of the amplitude, the highest value, and the area of the CoP ellipse, it was the 6-year-olds who achieved the lowest average values both when measured with eyes open and closed, so on this basis it can be concluded that they had the best control over the movement of the resultant foot support point and were characterized by the greatest stability during free standing. No statistically significant differences were observed in the static tests depending on the age of the sensory integration participants. The closer to 0.5 for each of the parameters in Table 1, the more even the load on the limbs. There were no statistically significant differences between the values of individual tests performed with eyes open and without eye control (i.e., eyes closed). The obtained results are presented in Table 1.

The greatest relationship between age and the value of a given parameter can be observed in the groups of 4- and 5-year-olds, because as many as 8 out of 12 measured parameters had a correlation equal to 1, and the next 2 parameters were also characterized by a value above 0.8. A similar situation concerned the groups of 5- and 6-year-olds, where we obtained a correlation equal to 1 for 7 out of 12 parameters, and the next 2 parameters also had a value above 0.8, which gives us information on the very high correlation between these compared groups. We obtained a full correlation in the smallest number of parameters in the comparison of groups of the 4- and 6-year-olds (4 out of 12 parameters). The above results are presented in Table 2.

In the study, before the start of therapy, statistically significant, positive high correlations were observed between the MCoCy_oe value and height and weight in 5-year-old children, while in the 4-year-olds the same measurement of balance was inversely related to body weight. In the group of 6-year-olds, a statistically significant positive high correlation between the SPL_oe and HoE_oe measurements and body height was observed. Other balance measurements did not show dependence on the height and weight of the examined children. There were no statistically significant correlations between body height and weight, and the results obtained in the measurement of static balance. The results are presented in Table 2.

After the SI program, statistically significant changes were observed in the values of MCoCy_oe, WoE_oe, and AoE_oe in the group of 4-year-olds. In the case of the MCoCy_oe parameter, its average values were higher than before therapy. However, in the case of the WoE_oe and AoE_oe parameters, the average values were lower, which indicates a decrease in the area and amplitude of the CoP point and, thus, an improvement in balance. In the group of 5-year-olds, a statistically significant change occurred in the case of the MCoCX_ce measurement and in the group of 6-year-olds in SPL_ce and AoE_ce. The obtained differences in the static balance measurements were not statistically significant. It is worth noting that in the case of the 4-year-olds and 6-year-olds, the average difference in the load on the right lower limb both with eyes open and with eyes closed had a negative sign, which means that this side was loaded more by them during free standing. There were no significant differences in the changes that were observed in individual trials regardless of the vision control in individual groups. The results are presented in Table 3.

There were no statistically significant differences in the changes in the values of balance and static balance measurements among the study groups.

A statistically significant, high positive correlation was observed between body height and changes in the SPL_oe, HoE_oe, and AoE_oe values in the group of 6-year-olds. In the group of 4-year-olds, a statistically significant correlation occurred only between body height and the change in the value of MCoCx_oe. In the group of 5-year-olds, a statistically significant high correlation was shown by the change in the SPL_oe value with body height. Statistically significant correlations in the 4- and 5-year groups were negative. In the study group, statistically significant relationships were observed between the change in static balance on the right side with eyes closed and body height only in the group of 6-year-olds. Other changes in static balance did not depend statistically significantly on the height and weight of the examined children. Detailed correlation results are presented in Table 4.

## 4. Discussion

Sensory integration is a developmental process based on the brain’s neuroplasticity [44,45,46]. The plasticity of a child’s brain at the age of 4–6 is diverse; their motor skills are largely dependent on external factors stimulating their development [47,48]. That is why it is so important to adapt external stimuli to a child’s abilities and age. However, in the case of children with reduced muscle tone, their motor development process deviates from the normative values for a given age. Does the use of SIT and, above all, the result obtained depend on age? The decrease in muscle tone that was observed in the study group was demonstrated, among others, by the lack of balance skills. Among the examined children, increased deflections in all planes were observed, which is indicated by an increase in the values of MCoCx_oe and MCo-Cy_oe. It is also important that these values increase with age and their highest values are in 5-year-olds, which may indicate a better age for starting SIT.

In this group of 5-year-old children, the least statistically significant changes in the values describing balance were observed. It is also worth noting that no differences were observed in the results for static balance and balance of subjects dependent on eye control. The lack of differences among the children aged 4–6 years old indicates a similar baseline condition in the examined patients. The lack of differences also indicates that children with reduced muscle tone require external therapeutic help and balance dysfunction does not disappear with development.

Evaluating the changes that occurred under the influence of the applied SIT, it can be concluded that in the study group positive changes could be noted in the group of 6-year-olds, especially if we take into account the balance tests. Although these changes were not statistically significant, they indicate a tendency to improve in balance after the application of the SIT. In the group of 4-year-olds, the changes obtained in the assessment of balance can be divided into two groups. The values of the path, ellipse amplitude, and highest value of the determined amplitude for CoP increased. Similar results were obtained by Miller et al., who conducted specialized training aimed at improving the central integration of afferent stimuli in patients with multiple sclerosis [39]. However, it should be remembered that the values obtained in studies on children and adults cannot be compared, rather only possible tendencies to changes can observed. The lack of possibility to compare children and adults was shown by the research of Peterson et al., who reported that only a child at the age of approximately 12 can be compared in terms of balance control [49]. In the case of the described group of children, balance regulation is still immature, which is particularly indicated by the values of measurements with eyes closed. Interestingly, both when measuring standing for 30 s with eyes open and eyes closed in the parameters of the CoP path along the horizontal and vertical axes (MCoCx and MCoCy) of the platform, no statistically significant differences were found in the 6-year-olds, while in the 5-year-olds there were differences along the horizontal axis of the platform, with eyes closed (MCoCx_ce); moreover, in the 4-year-olds with eyes open, significant tilts were noted relative to the vertical axis of the platform (MCoCy_oe). However, the values of the CoP deflection field decreased in each group, which indicates a positive result of the applied therapy.

In human development, especially in the progressive period, an important element of acquiring the ability to maintain balance is the stabilization of the center of gravity. In proper development, the load on the feet should be as close as possible to 50%, which proves that we load both the right and the left foot evenly. Deviations from these values indicate a shift of the center of gravity to one side. In the case of the examined children, a higher load on the right foot was observed both in the case of closed and open eyes. Even such a slight disproportion of the load may result in postural abnormalities later on. With the use of SIT in the examined children, a slight shift of the center of gravity to the left was noted, which results in equalizing the load on the feet and, thus, reduces the risk of late complications, such as postural defects. However, the observed changes were not statistically significant, which may be due to a too short a period of therapy.

The lack of significant differences between the changes, both in the value of balance and static balance, under the influence of the applied SIT in the examined children may indicate that in this age period the time of starting therapy is not significant. The changes that were observed do not correlate with age, but the height of the child’s body is more important. Especially in the group of 6-year-olds, a statistically significant high correlation can be observed between the height of the child’s body and the changes that occurred under the influence of SIT. It is also important that the correlated changes are positive, i.e., they indicate a positive effect of the therapy.

Surprisingly, the therapy was conducted for only two months, once a week, meaning that there were 9–10 meetings lasting 40 min, and the results obtained indicate improvement. What could have helped achieve such results? Sensory integration therapy consists of play-based classes during which children usually do not realize that they are “practicing” because they are having fun since the games and activities proposed by a therapist with a specific therapeutic goal. Therefore, it is worth considering the brain’s plasticity, which is responsible for the number of synapses. One nerve cell axon can create up to 15,000 synapses [50]. A baby is born with approximately twice as many synapses, as it needs to function, and over the next months and years after birth, the number is reduced according to the principle of “get rid of what you don’t use”. This process is explained by the fact that the brain of a small child functions like a “sponge”. It absorbs all the information that it hears, sees, and experiences—but if later it does not use certain words or do certain things, it will simply forget about them, i.e., the nerve impulses responsible for these experiences will cease to be generated, and, consequently, the synapses will cease to be used, i.e., they will disappear [50], which is a beneficial process, as those synapses that are used more often will be strengthened. For example, if a preschool child hears that combining blue and yellow paint results in green, but they will not test it or use this knowledge, they will forget about it. Suppose an improvement was noted after such a short duration and frequency of classes. In that case, it may mean that brain plasticity in children is very high and that properly conducted therapy can bring the desired effects.

Unfortunately, Sensory Integration is most often associated and addressed to children with autism, cerebral palsy or Down syndrome [5,42,51,52,53]. Its importance in terms of improving the use of the senses is emphasized. However, in each of these diseases, in addition to abnormalities in the perception of external stimuli, there are most often disturbances in the sense of one’s own body, including the sense of balance and spatial orientation. Limiting TSI to selected disease entities means that in the case of reduced muscle tone, it is not ordered as a therapeutic form. As can be seen from the conducted research, even short therapeutic sessions (one therapeutic session per week for a period of 10 weeks) leads to positive results, especially in the case of 5–6-year-old children.

In the literature on the subject, there are not too many items on SI aimed at improving body posture; this aspect is not assessed. The reason for such a situation may be the difficult study group, which consists of children in whom the diagnosis of reduced muscle tone is relatively difficult and very often delayed in time. Secondly, it is difficult to unify the SIT, because it is selected individually to the patient’s needs. However, the presented results of the study prove that sensory integration has a positive impact on the psychomotor development of children, and, therefore, research on the impact of sensory integration therapy on balance parameters and overall stability and development in children should be continued and deepened.

The research presented in this work was carried out on the previously mentioned ZEBRIS platform, which has a wide range of applications. One of the application options is the measurement of balance parameters in children with various health problems implemented in this work. It can also be used to observe healthy children, e.g., examining the relationship between balance and body mass composition in children and adolescents [54] or comparing the sway of the posture to the skin sensation of the sole of the foot depending on the frequency of saccadic eye movements (fast, associated eye movements eyes that move the image of the object from the peripheral part of the retina to its center) in young adults [55]. The group studied on the ZEBRIS platform can also be athletes, e.g., Sadowska D. et al. conducted a study on the impact of running phases on the posture balance of a modern pentathlete in laser running [56], as well as professional groups, e.g., Kasović M. et al. examined the impact of wearing police equipment on spatiotemporal and kinetic gait parameters in first-year police officers [57].

### Limitations

In terms of possible further research, the following should be taken into account: unification of the study group, because in the above-conducted experiment, the study group consisted of children with weakened axial muscle tone, but with different grounds: neurological, genetic, and caused by other injuries. In order for the research to be more reproducible, this group should be precisely defined and its size increased. In order to assess the long-term impact of SI on balance, the therapeutic process should be extended and follow-up examinations should be performed one year after the end of therapy. Another aspect that should be taken into account is the repetition of the examination at the age of 12, when there is a relative stabilization of body posture and the results of equivalent examinations are similar to those of adults. A major limitation for static balance and balance studies is the lack of standardized values adjusted to age and gender.

## 5. Conclusions

The sensory integration therapy used in the study group of 4–6-year-old children with reduced muscle tone led to positive results in the form of improved static balance and balance.

In the case of this age group, the age of the child at which SI is introduced is not important, but the body height, which positively correlates with the improvement of the values obtained in the equivalent tests on the ZEBRIS PDM-S platform, especially in the assessment of changes in the projection of the center of gravity.

## Figures and Tables

**Figure 1 children-10-00845-f001:**
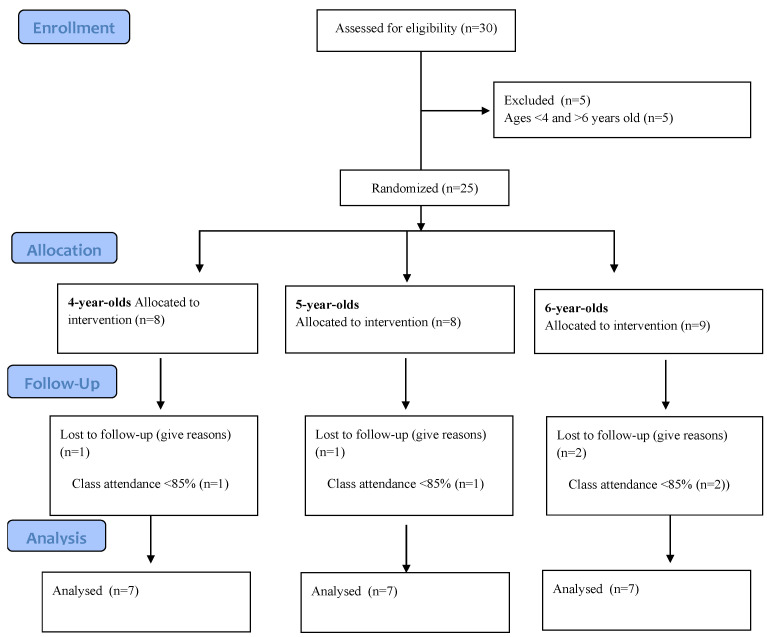
Qualification to the study group—CONSOR diagram.

**Table 1 children-10-00845-t001:** Comparison of the values of balance parameters and static measurement in children depending on the age before starting SI therapy.

Variable	4-Years-Olds (*n* = 7)	5-Years-Olds (*n* = 7)	6-Years-Olds (*n* = 7)	4 vs. 5	4 vs. 6	5 vs. 6
	Mean + Standard Deviation	Mean + Standard Deviation	Mean + Standard Deviation			
**1 MCoCx _oe (cm)**	17.12 ± 3.13	19.55 ± 3.88	19.03 ± 2.76	0.395	0.845	1.000
**1 MCoCx _ce (cm)**	17.56 ± 1.95	19.52 ± 2.67	19.18 ± 2.65	1.000	1.000	1.000
*p*		0.892	0.893			
**1 MCoCy_oe (cm)**	21.38 ± 3.01	26.80 ± 2.63	24.16 ± 3.40	0.010 *	0.255	0.683
**1 MCoCy_ce (cm)**	22.35 ± 4.86	23.09 ± 3.80	23.53 ± 2.53	1.000	1.000	1.000
*p*	0.715	0.138	0.224			
**1 SPL _oe (cm)**	222.10 ± 87.35	244.63 ± 122.24	241.72 ± 223.33	1.000	1.000	0.966
**1 SPL_ce (cm)**	427.99 ± 333.85	228.02 ± 87.16	198.49 ± 89.20	1.000	0.510	0.870
*p*	0.465	0.892	0.685			
**1 WoE_oe (cm)**	17.08 ± 13.12	16.26 ± 11.75	7.80 ± 6.96	1.000	0.305	0.333
**1 WoE_ce (cm)**	14.98 ± 10.33	9.98 ± 8.46	7.81 ± 6.68	1.000	0.544	1.000
*p*	0.678	0.500	0.500			
**1 HoE_oe (cm)**	11.69 ± 6.53	8.69 ± 3.50	6.89 ± 4.77	1.000	0.363	1.000
**1 HoE_ce (cm)**	21.66 ± 18.09	8.22 ± 3.37	11.77 ± 11.36	0.904	1.000	1.000
*p*	0.273	0.345	0.345			
**1 AoE_oe (cm^2^)**	207.83 ± 237.51	123.82 ± 106.42	49.47 ± 69.00	1.000	0.305	0.305
**1 AoE_ce (cm^2^)**	353.37 ± 430.49	82.36 ± 97.82	51.11 ± 41.82	0.879	0.698	1.000
*p*	0.465	0.500	0.685			
**1 load on the left side_oe**	0.477 ± 0.171	0.467 ± 0.164.	0.476 ± 0.069	1.000	1.000	1.000

* Statistically significant differences for *p* < 0.05.

**Table 2 children-10-00845-t002:** Correlation of balance parameters and static measurements of age and body weight in the examined children.

Variable	4-Years-Olds		5-Years-Olds		6-Years-Olds	
	Growth (cm)	Body Weight (kg)	Growth (cm)	Body Weight (kg)	Growth (cm)	Body Weight (kg)
**1 MCoCx _oe (cm)**	−0.505	−0.394	0.408	−0.418	−0.187	0.037
**1 MCoCx _ce (cm)**	−0.105	−0.258	−0.154	0.359	−0.112	0.224
**1 MCoCy_oe (cm)**	−0.518	−0.915 *	0.815 *	0.782 *	−0.337	0.704
**1 MCoCy_ce (cm)**	0.211	−0.775	0.154	0.051	−0.112	0.224
**1 SPL _oe (cm)**	−0.072	0.394	−0.296	0.091	0.879 *	−0.334
**1 SPL_ce (cm)**	−0.211	0.775	−0.667	−0.154	0.671	−0.224
**1 WoE_oe (cm)**	0.072	0.532	0.222	0.327	0.505	−0.222
**1 WoE_ce (cm)**	−0.211	0.775	0.051	0.359	0.112	−0.224
**1 HoE_oe (cm)**	0.090	0.571	0.000	0.327	0.879 *	−0.556
**1 HoE_ce (cm)**	−0.211	0.775	0.051	0.359	0.112	−0.671
**1 AoE_oe (cm^2^)**	−0.090	0.433	0.222	0.327	0.692	−0.334
**1 AoE_ce (cm^2^)**	−0.211	0.775	0.051	0.359	0.447	−0.447
**1 Load on the left side_oe**	−0.234	−0.197	0.185	0.218	0.038	−0.299
**1 Load on the right side_oe**	0.144	0.197	−0.185	−0.218	−0.038	0.299
**1 Load on the left side_ce**	−0.211	0.775	0.051	0.359	−0.335	−0.224
**1 Load on the right side_oe**	0.211	−0.775	−0.051	−0.359	0.335	0.224

* Correlations statistically significant.

**Table 3 children-10-00845-t003:** The difference in the average value of balance parameters and static measurement in children depending on their age before and after sensory integration therapy.

Variable	4-Years-Olds		5-Years-Olds		6-Years-Olds	
	Difference Mean + SD	*p*	Difference Mean + SD	*p*	Difference Mean + SD	*p*
**1-2 MCoCx _oe (cm)**	−2.60 ± 3.44	0.063	1.63 ± 5.07	0.310	0.14 ± 3.31	0.735
**1-2 MCoCx _ce (cm)**	−0.85 ± 3.16	0.715	1.41 ± 0,51	0.043 *	2.56 ± 3.33	0.138
*p*	0.465		0.893		0.138	
**1-2 MCoCy_oe (cm)**	−14.46 ± 27.73	0.028 *	1.11 ± 4.02	0.735	0.90 ± 5.53	0.499
**1-2 MCoCy_ce (cm)**	−2.09 ± 6.1	0.465	−3.44 ± 4.43	0.225	−0.86 ± 4.36	0.500
*p*	1.000		0.138		0.892	
**1-2 SPL _oe (cm)**	48.15 ± 56.58	0.063	39.25 ± 123.24	0.866	112.50 ± 222.27	0.176
**1-2 SPL_ce (cm)**	123.69 ± 184.90	0.273	54.24 ± 73.51	0.225	71.08 ± 104.43	0.043 *
*p*	0.273		0.892		0.892	
**1-2 WoE_oe (cm)**	9.50 ± 11.69	0.043 *	6.06 ± 6.79	0.063	3.67 ± 6.81	0.063
**1-2 WoE_ce (cm)**	6.53 ± 6.18	0.068	4.76 ± 8.90	0.686	3.84 ± 5.58	0.138
*p*	1.000		0.138		0.892	
**1-2 HoE_oe (cm)**	4.89 ± 6.76	0.063	0.14 ± 5.58	0.735	2.14 ± 4.53	0.499
**1-2 HoE_ce (cm)**	4.95 ± 13.44	0.465	0.81 ± 2.65	0.500	5.34 ± 11.89	0.500
*p*	0.715		0.685		0.685	
**1-2 AoE_oe (cm^2^)**	171.30 ± 230.01	0.018 *	17.86 ± 130.48	0.176	34.61 ± 67.92	0.176
**1-2 AoE_ce (cm^2^)**	176.29 ± 256.09	0.273	51.20 ± 95.46	0.686	31.53 ± 31.57	0.043 *
*p*	0.465		0.345		0.685	
**1-2 Load on the left side_oe**	−0.026 ± 0.146	0.735	0.013 ± 0.122	0.866	−0.023 ± 0.080	0.345
**1-2 Load on the left side_ce**	−0.005 ± 0.111	1.000	0.000 ± 0.082	0.686	−0.082 ± 0.127	0.225
*p*	0.715		0.685		0.225	
**1-2 Load on the right side_oe**	0.031 ± 0.138	0.612	−0.013 ± 0.122	0.866	0.023 ± 0.080	0.345
**1-2 Load on the right side_ce**	0.005 ± 0.111	1.000	0.000 ± 0.082	0.686	0.082 ± 0.127	0.225
*p*	0.855		0.685		0.225	

* Statistically significant changes.

**Table 4 children-10-00845-t004:** Relationship between changes in balance and static measurements and body weight and height depending on the age of the examined children.

Variable	4-Years-Olds		5-Years-Olds		6-Years-Olds	
	Growth (cm)	Body Weight (kg)	Growth (cm)	Body Weight (kg)	Growth (cm)	Body Weight (kg)
**1-2 MCoCx _oe (cm)**	−0.829 *	−0.611	0.593	−0.255	−0.187	−0.408
**1-2 MCoCx _ce (cm)**	−0.738	0.775	−0.616	−0.205	−0.335	−0.224
**1-2 MCoCy_oe (cm)**	−0.378	−0.039	0.482	0.727	−0.281	0.408
**1-2 MCoCy_ce (cm)**	−0.105	−0.258	0.205	0.616	−0.112	0.224
**1-2 SPL _oe (cm)**	0.487	0.473	−0.778 *	−0.218	0.767 *	−0.259
**1-2 SPL_ce (cm)**	0.632	0.258	−0.667	−0.154	0.335	0.224
**1-2 WoE_oe (cm)**	0.000	0.197	0.185	−0.073	0.692	−0.334
**1-2 WoE_ce (cm)**	0.632	0.258	0.051	0.359	0.112	−0.224
**1-2 HoE_oe (cm)**	0.162	0.512	−0.482	−0.145	0.879 *	−0.556
**1-2 HoE_ce (cm)**	0.632	0.258	−0.667	−0.154	0.447	0.671
**1-2 AoE_oe (cm^2^)**	0.198	0.611	−0.334	0.127	0.879 *	−0.334
**1-2 AoE_ce (cm^2^)**	0.105	0.258	−0.667	−0.154	0.447	−0.447
**1-2 Load on the left side_oe**	−0.072	−0.099	−0.580	0.009	0.019	0.150
**1-2 Load on the right side_oe**	−0.018	0.099	0.580	−0.009	−0.019	−0.150
**1-2 Load on the left side_ce**	−0.211	0.775	0.237	−0.342	−0.803	−0.344
**1-2 Load on the right side_oe**	0.211	−0.775	−0.237	0.342	0.894 *	0.447

* Statistically significant correlations.

## Data Availability

The data that support the findings of this study are available from the corresponding author upon request.

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
