# Peer review of "Assessment of Balance Parameters in Children with Weakened Axial Muscle Tone Undergoing Sensory Integration Therapy"

_children, 2023, doi:10.3390/children10050845_

Round 1

Reviewer 1 Report

Thank you for sending the manuscript again, it has improved a lot but I think there are aspects to improve.

Abstract

- This text cannot go in the first part of the abstract, since it is a result: "The study group consisted of 21 children (four, five 12 and six years old) with weakened axial muscle tone" 

- In methods, it should be stated that an analysis has been carried out by age groups.

Introduction

- This phrase at the beginning is not understood very well: "Recent The development of a child, although determined by certain standards adapted to age, depends on the environment in which the child is brought up as well as on his health."

- Children with normal development present a normal SI, because if there is involvement of a sensory channel and its integration, it will not be typical.

Methods

- Again, this text goes to results: "The study group consisted of 21 children aged 4-6 100 years. The subjects were divided into three groups according to age - group I (n=7) of 101 children aged 4, group II (n=7) - 5-year-olds, group III (n=7) - 6-year-olds" 

- Consor's Figure to results and make it look good, the arrows have been moved

Results

- Much better results like this in terms of tables, but I think that in the section everything could be put together by age and thus facilitate the reading of the text

Discussion

- You have to give the results of the study in the first paragraph so that you can then discuss the most interesting results. This part should be restructured, so that with each finding a discussion can be made based on the studies presented.

Conclusions

OK

Author Response

Thank you for sending the manuscript again, it has improved a lot but I think there are aspects to improve.

Thank you very much for your constructive comments. I hope that our answers to your comments will be satisfactory and will allow for further evaluation of the work.

Abstract

- This text cannot go in the first part of the abstract, since it is a result: "The study group consisted of 21 children (four, five 12 and six years old) with weakened axial muscle tone" 

- In methods, it should be stated that an analysis has been carried out by age groups.

 Information on the division into three age groups has been improved:

The study group consisted of 21 children (divided into three age groups) referred by a doctor for therapy.

Introduction

- This phrase at the beginning is not understood very well: "Recent The development of a child, although determined by certain standards adapted to age, depends on the environment in which the child is brought up as well as on his health."

Corrected

- Children with normal development present a normal SI, because if there is involvement of a sensory channel and its integration, it will not be typical.

 Corrected

Methods

- Again, this text goes to results: "The study group consisted of 21 children aged 4-6 100 years. The subjects were divided into three groups according to age - group I (n=7) of 101 children aged 4, group II (n=7) - 5-year-olds, group III (n=7) - 6-year-olds" 

- Consor's Figure to results and make it look good, the arrows have been moved

 Corrected

Results

- Much better results like this in terms of tables, but I think that in the section everything could be put together by age and thus facilitate the reading of the text

Thank you very much for your opinion, all results are broken down by age. Each age group is in a separate column. We tried to build the tables so that they were legible and there were not too many of them, so as not to duplicate the parameters in each table separately.

Discussion

- You have to give the results of the study in the first paragraph so that you can then discuss the most interesting results. This part should be restructured, so that with each finding a discussion can be made based on the studies presented.

Thank you for your feedback. The discussion has been restructured and changed based on your suggestion.

Conclusions

OK

Thank you for your commitment, we hope that the changes introduced are transparent and the answers are reliable and our article will be published. If you have any questions, please contact me.

Kind regards,

Jadwiga Jacewicz

Reviewer 2 Report

Dear authors,

it is the second time that I find myself reviewing this manuscript and, although it is much better than the last version, I still notice fundamental methodological aspects that need to be improved. For instance, considering the age of the subgroups in the sample and the duration of the therapeutic intervention, in order to be able to affirm that the results obtained are actually attributable to the proposed intervention and not to a physiological developmental effect linked to growth, it would have been important to insert the analysis of a comparable cohort that did not follow the same therapeutic path. Furthermore, it would be advisable to specify the cause of the reduction in muscle tone and to verify whether the presumed improvements are independent or not from the pathological condition that led to the its reduction.

Also, I would add the following minors:

- Line 53: "The current goal of Sensory Integration Therapy (SIT) is to reduce or eliminate problems of neuronal disorganization." Could you explain the concept of neural disorganization more clearly?

- what is the sampling frequency of the ZEBRIS PDM-S Platform?

- Line 287: "It is also important that these values ​​increase with age and their highest values ​​are in 5-year-olds, which may indicate a borderline age for starting SIT." Speculation not supported by the data, given the smallness of the sample.

Author Response

Thank you very much for your constructive comments. I hope that our answers to your comments will be satisfactory and will allow for further evaluation of the work.

Dear authors,

it is the second time that I find myself reviewing this manuscript and, although it is much better than the last version, I still notice fundamental methodological aspects that need to be improved. For instance, considering the age of the subgroups in the sample and the duration of the therapeutic intervention, in order to be able to affirm that the results obtained are actually attributable to the proposed intervention and not to a physiological developmental effect linked to growth, it would have been important to insert the analysis of a comparable cohort that did not follow the same therapeutic path. Furthermore, it would be advisable to specify the cause of the reduction in muscle tone and to verify whether the presumed improvements are independent or not from the pathological condition that led to the its reduction.

The presented research was carried out in a therapeutic center to which children diagnosed with motor development problems are referred. It is not possible to test children who do not have balance problems at the same time. We understand that the presented period is a very dynamic period in the child's ontogenetic development, however, AI is used by children whose development is delayed and require stimulation. If they were compared to their peers, their results would be much worse and might not indicate an improvement under the influence of therapy, as they would still be outside the normal range. Unfortunately, there are no normative values for children's balance tests, so the results were compared at a very similar age for 4-, 5- and 6-year-olds. There are very few publications in the literature on the improvement of balance in children, therefore it is difficult to compare such studies.

At the same time, it provides a basis for further research, and the reviewer's comments will certainly be taken into account in subsequent studies.

Also, I would add the following minors:

- Line 53: "The current goal of Sensory Integration Therapy (SIT) is to reduce or eliminate problems of neuronal disorganization." Could you explain the concept of neural disorganization more clearly?

Neuronal disorganization means a problem in responding appropriately to the stimuli reaching the body. As a result of the fact that the senses in our body may incorrectly receive or process the stimuli that reach us, there are difficulties in the field of sensory integration, which are worked on during SIT.

- what is the sampling frequency of the ZEBRIS PDM-S Platform?

The information has been supplemented

- Line 287: "It is also important that these values ​​increase with age and their highest values ​​are in 5-year-olds, which may indicate a borderline age for starting SIT." Speculation not supported by the data, given the smallness of the sample.

Corrected

Thank you for your commitment, we hope that the changes introduced are transparent and the answers are reliable and our article will be published. If you have any questions, please contact me.

Kind regards,

Jadwiga Jacewicz

Reviewer 3 Report

Thank you for the opportunity to review this interesting article on improving balance through specific modern therapies.

My recommendations are included in the text of the manual and attached to this message.

Good luck.

Author Response

Thank you very much for your constructive comments. I hope that our answers to your comments will be satisfactory and will allow for further evaluation of the work. Our answears are attached. 

Kind regards,

Jadwiga Jacewicz

Reviewer 4 Report

Dear authors,

Thank you very much for your contribution, I consider it may be very suitable for publication in Children.

Although, I believe that a series of considerations should be taken into account:

Introduction

-       Line 29, delete Recent

-       Line 31, if it is possible I will include a citation after “his health”

-       Line 48, join the hyphen with the 7

-       Line 68, join the comma [26,33]

-       I think it is important to include a Hypothesis at the end of introduction.

Materials and methods

-       Split the section including Study design (quasiexperimental study…. With reference); Process; Statistical analysis

-       Do the authors think that if they had participated more than once a week they could have improved the results? How long did the intervention sessions last?

-       Take care of the white space during the manuscript (for example line 123 and 125 or 337 with a double space)

Discussion

-       I consider it important to start it with the hypothesis and purpose of the study

-       Split the discussion into different paragraphs (the second one is too long)

-       Why do the authors think that the results are different in the variables according to age? (for instance, why the changes to AoE_oe was only in the six-years old group)

-       The discussion is clear but too long, I think it should be summarized by including the references. For example, lines 268 to 304 do not have bibliographic references, this must be summarized and modified. Same lines 308-344. The discussion must be modified so that the bibliographic foundation is seen, not "only the opinion of the authors"

-       Include “future prospects”

References

-       Review the rules of MDPI (for example, blonde number in publication year and italics in the journal and the number. The abbreviated title of the journals should be included)

Author Response

Thank you very much for your constructive comments. I hope that our answers to your comments will be satisfactory and will allow for further evaluation of the work.

Dear authors,

Thank you very much for your contribution, I consider it may be very suitable for publication in Children.

 Although, I believe that a series of considerations should be taken into account:

Introduction

-       Line 29, delete Recent

Removed

-       Line 31, if it is possible I will include a citation after “his health”

The citation for this passage is contained in line 36 [1-4]

-       Line 48, join the hyphen with the 7

Corrected

-       Line 68, join the comma [26,33]

Corrected

-       I think it is important to include a Hypothesis at the end of introduction

At the end of the introduction, research questions were posed, which in the case of such a small group and additionally pilot studies give the opportunity to obtain answers and direct further research. The hypothesis in such a small research group is easy to undermine. The aim of the research was supplemented in the work

Materials and methods

-       Split the section including Study design (quasiexperimental study…. With reference); Process; Statistical analysis

Corrected

-       Do the authors think that if they had participated more than once a week they could have improved the results? How long did the intervention sessions last?

As written, each session was 40 minutes. – line 125

The frequency of classes depended on the organizational possibilities of the parents. At the same time, the planning of classes was in line with the recommendations of the referring physician. Children participating in the study were not in a therapeutic center - the form of classes was outpatient

However, we thank you for this attention and in the future we will offer parents more frequent visits

-       Take care of the white space during the manuscript (for example line 123 and 125 or 337 with a double space)

 Corrected

 Discussion

-       I consider it important to start it with the hypothesis and purpose of the study

Corrected

-       Split the discussion into different paragraphs (the second one is too long)

Corrected

-       Why do the authors think that the results are different in the variables according to age? (for instance, why the changes to AoE_oe was only in the six-years old group)

All the results are presented in all groups. The obtained results are differentiated by age, the differences are not always statistically significant, but they do occur

-       The discussion is clear but too long, I think it should be summarized by including the references. For example, lines 268 to 304 do not have bibliographic references, this must be summarized and modified. Same lines 308-344. The discussion must be modified so that the bibliographic foundation is seen, not "only the opinion of the authors"

The discussion was corrected

-       Include “future prospects”

Future elements are included in the Limitation subsection

 References

-       Review the rules of MDPI (for example, blonde number in publication year and italics in the journal and the number. The abbreviated title of the journals should be included)

Corrected

Thank you for your commitment, we hope that the changes introduced are transparent and the answers are reliable and our article will be published. If you have any questions, please contact me.

Kind regards,

Jadwiga Jacewicz

Round 2

Reviewer 2 Report

Dear authors,

i'm afraid i wasn't clear enough in the last revision. When I stated that in order to be able to argue that the results obtained are attributable to the proposed intervention and not to an evolutionary effect linked to growth it would have been important to include the analysis of a comparable cohort that did not follow the same therapeutic path, i was certainly not referring to a group of healthy subjects. I was referring to a group of subjects with the same pathology (whatever it is), in order to verify that the improvements related to balance were plausibly due to the proposed protocol and not to an effect linked to growth. I believe that the issue is very interesting and that it is certainly something that must be explored. However, as proposed to date, for the aforementioned reasons, i think the results cannot support beyond any reasonable doubt that the proposed intervention works. Also, I would again recommend more clarity on the disease state leading to hypotonia in order to be able to propose a plausible explanation as to why this specific therapy is useful in the specific pathological context.

Author Response

Thank you very much for your constructive comments. I hope that our answers to your comments will be satisfactory and will allow for further evaluation of the work.

Dear authors,

I'm afraid I wasn't clear enough in the last revision. When I stated that in order to be able to argue that the results obtained are attributable to the proposed intervention and not to an evolutionary effect linked to growth it would have been important to include the analysis of a comparable cohort that did not follow the same therapeutic path, i was certainly not referring to a group of healthy subjects. I was referring to a group of subjects with the same pathology (whatever it is), in order to verify that the improvements related to balance were plausibly due to the proposed protocol and not to an effect linked to growth. I believe that the issue is very interesting and that it is certainly something that must be explored. However, as proposed to date, for the aforementioned reasons, i think the results cannot support beyond any reasonable doubt that the proposed intervention works. Also, I would again recommend more clarity on the disease state leading to hypotonia in order to be able to propose a plausible explanation as to why this specific therapy is useful in the specific pathological context.

The primary purpose of the presented study was not to assess the effectiveness of the SI compared to another method, but to examine whether the age at which we use this method is significant. Therefore, no other therapeutic form was included in the studies. We wanted to answer the question whether age-related motor development will significantly affect the results obtained by the examined children after using the SI. Therefore, in the studies we compared the changes that occurred under the influence of the therapeutic program, and not the result obtained after the therapy.

Another reason why we did not examine children treated with a different form of therapy was the nature of the center where the research was conducted. Children with weakened axial muscle tone started the therapeutic program with SI classes. Introducing a different therapeutic form to the research would result in heterogeneity of the group in terms of basic diagnosis or duration of therapy. In the study group, children were referred for therapy for the first time after the diagnosis. In addition, the research was carried out in 2021, i.e. during the period of restrictions related to the pandemic caused by the Sars-Cov-2 virus, which was associated with limited access to physiotherapy services and a reduced possibility of undertaking other activities by the tested children.

The current control group would consist of children who do not have restrictions in the form of restrictions, who can participate in games in kindergarten or on the playground. Of course, in future work, we will take into account the reviewer's suggestion and try to study a control group.

Thank you for your commitment, we hope that the answers are reliable and our article will be published. If you have any questions, please contact me.

Kind regards,

Jadwiga Jacewicz

Reviewer 3 Report

Te autors improved the manuscript according with the recommendations. 

Author Response

Thank you again for all constructive feedback. We hope that after significant corrections our article will be published.

Yours faithfully,

Jadwiga Jacewicz

Round 3

Reviewer 2 Report

Dear authors,

i have not suggested a comparison between the therapeutic protocol used and another therapeutic protocol; my suggestion was (and is) to compare the data you obtained with those of a cohort of subjects comparable in terms of age and pathology who do not undergo any specific therapy. This  should be done in order to evaluate whether in 2 months (period of duration of the protocol) there are no changes in the balance parameters even in subjects who do not follow the therapeutic protocol due to a physiological growth effect. After all, you  have admitted to be aware that the period presented is a very dynamic period in the ontogenetic development of the child. This analysis would allow you to support the effectiveness of the SI protocol with greater credibility; advice is to add the proposed comparison.

Author Response

Thank you very much for your constructive comments. I hope that our answers to your comments will be satisfactory and will allow for further evaluation of the work.

Dear authors,

I have not suggested a comparison between the therapeutic protocol used and another therapeutic protocol; my suggestion was (and is) to compare the data you obtained with those of a cohort of subjects comparable in terms of age and pathology who do not undergo any specific therapy. This  should be done in order to evaluate whether in 2 months (period of duration of the protocol) there are no changes in the balance parameters even in subjects who do not follow the therapeutic protocol due to a physiological growth effect. After all, you  have admitted to be aware that the period presented is a very dynamic period in the ontogenetic development of the child. This analysis would allow you to support the effectiveness of the SI protocol with greater credibility; advice is to add the proposed comparison.

Dear Reviewer

Thank you very much for your comments. The problem of comparing our research with the control group is difficult to consider. The research presented by us was carried out in a rather specific period - restrictions related to the Covid-19 pandemic. In Poland, it was a period of closed centers for children, schools, kindergartens, limited access to sports centers. For the examined children - conducted at Si's was the only form of therapy apart from what the children had as standard at home. At the same time, the balance and balance studies we conducted are not standard studies on healthy children, which is why it is difficult to make comparisons. ONLY children referred for therapy come to the center, and in kindergartens diagnostics for reduced muscle tone are not carried out. The main aim of our study was to assess the difference that can occur in 4-year-old, 5-year-old and 6-year-old children stimulated with SI. As our research showed, the improvement of balance and balance was influenced by body height, not age, which gives grounds for further research. We will definitely include a control group for further research. However, during the Covid-19 pandemic, it was not possible to perform comparative tests, and supplementing the control group now from a methodological point of view is incorrect because there are no such restrictions in terms of physical activity as there were in the research period presented by us. The most important thing is that the differences that were observed in individual measurements do not differ depending on the age group, which may indicate the impact of SI therapy and not a developmental leap on the improvement of these values

Thank you for your commitment, we hope that the changes introduced are transparent and the answers are reliable and our article will be published. If you have any questions, please contact me.

Kind regards,

Jadwiga Jacewicz
